# Isocitrate Dehydrogenase Inhibitors in Glioma: From Bench to Bedside

**DOI:** 10.3390/ph17060682

**Published:** 2024-05-26

**Authors:** Merve Hazal Ser, Mason Webb, Anna Thomsen, Ugur Sener

**Affiliations:** 1Department of Neurology, SBU Istanbul Research and Training Hospital, Istanbul 34098, Turkey; 2Department of Medical Oncology, Mayo Clinic, Rochester, MN 55905, USA; webb.mason@mayo.edu (M.W.); sener.ugur@mayo.edu (U.S.); 3Department of Neurology, Mayo Clinic, Rochester, MN 55905, USA

**Keywords:** glial tumor, IDH inhibitor, vorasidenib, dual inhibitor of mIDH1/2

## Abstract

Isocitrate dehydrogenase (IDH) mutant gliomas are a primary malignancy of the central nervous system (CNS) malignancies, most commonly affecting adults under the age of 55. Standard of care therapy for IDH-mutant gliomas involves maximal safe resection, radiotherapy, and chemotherapy. However, despite good initial responses to multimodality treatment, recurrence is virtually universal. IDH-mutant gliomas represent a life-limiting prognosis. For this reason, there is a great need for novel treatments that can prolong survival. Uniquely for IDH-mutant gliomas, the IDH mutation is the direct driver of oncogenesis through its oncometabolite 2-hydroxygluterate. Inhibition of this mutated IDH with a corresponding reduction in 2-hydroxygluterate offers an attractive treatment target. Researchers have tested several IDH inhibitors in glioma through preclinical and early clinical trials. A phase III clinical trial of an IDH1 and IDH2 inhibitor vorasidenib yielded promising results among patients with low-grade IDH-mutant gliomas who had undergone initial surgery and no radiation or chemotherapy. However, many questions remain regarding optimal use of IDH inhibitors in clinical practice. In this review, we discuss the importance of IDH mutations in oncogenesis of adult-type diffuse gliomas and current evidence supporting the use of IDH inhibitors as therapeutic agents for glioma treatment. We also examine unresolved questions and propose potential directions for future research.

## 1. Introduction

Gliomas represent the most common primary brain malignancies in adults [1]. As our understanding of tumor biology improves, the classification of adult-type gliomas has increasingly included molecular features such as the presence of isocitrate dehydrogenase (IDH) mutations or the co-occurring presence of whole-arm deletions in chromosome 1p and 19q [2,3,4,5]. Accordingly, the World Health Organization Classification of Tumors of the Central Nervous System (WHO CNS) 2021 separates adult-type diffuse gliomas into three groups, defined as astrocytoma, IDH-mutant; oligodendroglioma, IDH-mutant and 1p/19q co-deleted; and glioblastoma, IDH-wildtype [6]. Historically, IDH-mutant adult-type diffuse gliomas have been characterized by more indolent behavior and a better response to conventional therapies when compared to IDH-wildtype tumors [7].

Standard of care therapy for IDH-mutant gliomas involves maximal safe resection, radiotherapy, and chemotherapy, either with temozolomide or a combination of procarbazine, lomustine, and vincristine (PCV) [8]. Patients are classically stratified into high-risk or low-risk categories based on a variety of factors including tumor grade, patient age at diagnosis, and the extent of resection achieved [9]. Patients with grade 2 IDH-mutant astrocytomas or oligodendrogliomas who are younger than 40 and receive a gross total resection are often managed with expectant surveillance until tumor recurrence [10]. Otherwise, patients who are older than 40, have had a subtotal resection, or those with higher grade tumors are typically managed with irradiation and chemotherapy [9]. Systemic therapy with either temozolomide or PCV are acceptable [11,12,13]. However, despite initial good responses to multimodality treatment, recurrence is universal and IDH-mutant gliomas are life-limiting malignancies. Additionally, treatment with radiation therapy and conventional chemotherapy is associated with potentially serious toxicities including cognitive decline, accelerated atherosclerosis, fatigue, weight loss, liver injury, and myelosuppression. There is a great need for novel treatments that can prolong survival and mitigate the toxicities associated with conventional approaches.

In this review, we discuss the role of IDH mutations in the oncogenesis of adult-type diffuse gliomas and the current evidence supporting the use of IDH inhibitors as therapeutic agents for glioma treatment. We also discuss unresolved questions and potential directions for future research.

## 2. The Biological Effects of IDH Mutations on Gliomagenesis

The IDH enzyme family is made up of three isoforms which play a critical role in cellular metabolism. IDH1 is located in the cytoplasm and peroxisome, whereas IDH2 and IDH3 are located in the mitochondria [14]. IDH1 and IDH2 mutations have been recognized as oncogenic drivers [15]. In the mitochondria, isocitrate molecules undergo catalysis by nicotinamide adenine dinucleotide phosphate (NADP)-dependent IDH2, producing NADPH (the reduced form of NADP) and α-ketoglutarate (αKG).

IDH mutations result in a gain of neomorphic function, with the mutated enzyme demonstrating a decreased affinity for isocitrate and an increased affinity for αKG [16,17,18]. Consequently, this leads to the production of the oncometabolite D-2-hydroxygluterate (2HG) from αKG in a NADPH-dependent manner [19]. D-2-hydroxygluterate functions as an oncometabolite by influencing cellular metabolism, epigenetic modification, redox regulation, and DNA repair.

Given that αKG serves as the substrate for glycolysis in the Krebs cycle, the oncometabolite 2HG subsequently leads to a reduction in glycolysis by depleting αKG. To compensate, non-Krebs cycle sources of carbohydrates are needed. This compensation is achieved by glutaminolysis, which is a key mechanism maintaining metabolic balance by restoring αKG levels [20,21]. The consumption of NADPH by mutated IDH undermines de novo lipogenesis. Glutamine-driven lipogenesis also fulfills the need for lipid production, along with exogenous lipid sources [22]. Furthermore, lactate dehydrogenase activity is suppressed as the promoter region of its gene is silenced in response to 2HG, resulting in diminished rates of glycolysis [23].

Neomorphic IDH activity results in epigenetic reprogramming through both global DNA hypermethylation and histone methylation. DNA hypermethylation is driven by 2HG, which blocks the activity of the DNA demethylation enzyme, and a glioma-specific DNA methylation pattern occurs over time, leading to transcriptional silencing of tumor suppressor genes [24]. Ten-eleven translocation (TET) 5′ methylcytosine hydroxylase enzyme activity, which catalyzes a key step in the removal of DNA methylation, is blocked by 2HG. Thus, IDH mutations manifest a CpG island methylator phenotype (G-CIMP), leading gene expression programs of IDH-mutant glioma [25]. In addition, IDH mutations cause hypermethylation of CCCTC binding factor (CTCF) binding sites genome-wide, leading to reduced CTCF binding [26]. CTCF insulator protein is responsible for the partition of “contact domains” of the genome, which are discrete structural and regulatory units. Loss of CTCF at a domain boundary is linked to altered gene insulations, which contribute to gliomagenesis by activating oncogenes that are normally insulated. In mouse oligodendrocyte progenitor cells, disruption of CTCF insulator near the platelet-derived growth factor receptor A (PDGFRA), a prominent glioma oncogene, has been demonstrated to be related to increased proliferation [27]. Histone lysine demethylases (KDMs) are also inhibited by 2HG, which compromises cellular differentiation and contributes to the glioma cells’ regression to a more “primitive” developmental state [28]. Histone methylation by KDM inhibiton contributes to oncogenesis in IDH-mutant tumors through glial transformation in IDH-mutant cells [29,30].

The presence of IDH mutation also leads to the accumulation of reactive oxygen species (ROS), which are detrimental to DNA, lipids, and proteins. The mutated IDH depletes cellular NADPH, which impairs important activities that remove ROS, such as reducing glutathione disulfide [31,32]. Therefore, the accumulation of oxidative stress is a characteristic feature of cancer biology in IDH mutated tumors [33]. Glioma cells counterbalance excessive oxidative stress by increased production of manganese superoxide dismutase and protein carbonylation, which contributes to oncogenic transformation through genomic instability, loss of growth control, and invasiveness [34,35].

Overall, IDH mutations are a direct driver of oncogenesis in IDH-mutant glioma through the oncometabolite 2HG (Figure 1). Such IDH mutations have been demonstrated in a variety of cancers including glioma, cholangiocarcinoma, acute myeloid leukemia, and chondrosarcoma [36,37,38,39,40,41]. Given the central role of IDH mutation in gliomagenesis, inhibition of mutated IDH with resultant reduction in 2HG represents an attractive treatment target.

## 3. IDH Inhibitors in Glioma

### 3.1. Preclinical Evidence and Early Phase Studies

IDH inhibitors have been studied in a variety of human cancers, including glioma [2,4]. In 2013, Rohle et al. reported on the use of AGI-5198, a potent inhibitor of the IDH1 R132 mutated homodimer in an oligodendroglioma cell line [42]. AGI-5198 administration suppressed tumor cell growth in both low and high dosages (150 mg/kg and 450 mg/kg). At the higher dosage, 2HG reduction was accompanied by gliogenic differentiation, evidenced by increased RNA levels of astrocytic markers such as glial fibrillary acidic protein (GFAP) and aquaporin 4, along with decreased RNA levels of nestin, a marker for undifferentiated neuroprogenitor cells [42]. Importantly, these RNA expression changes did not correlate with DNA methylation changes. Therefore, the authors concluded that inhibiting mutated IDH would lead to impaired glioma growth even in the absence of DNA methylation. This suggested that mutated IDH promotes glioma growth through epigentic-independent mechanisms. Similar results were obtained with AGI-6780, an IDH2 inhibitor, which induced cellular differentiation in human leukemia cells, and an IDH1 inhibitor, which restored cellular differentiation of IDH1-mutant mouse hepatoblasts [43,44]. Clinical studies evaluating enasidenib, an IDH2 inhibitor, and ivosidenib, an IDH1 inhibitor, in IDH-mutant acute myeloid leukemia (AML) patients with recurrent or refractory disease showed durable remission, with some patients achieving full remission [45,46]. Similarly, ivosidenib was demonstrated to be related to improved progression-free survival (PFS) and overall survival (OS) in cholangiocarcinoma patients with the IDH1 mutation [38,47]. The FDA approved enasidenib and ivosidenib for IDH-mutant AML in 2017 and 2018, respectively, and ivosidenib for IDH-mutant cholangiocarcinoma in 2021 [48,49,50]. Significant progress has been made in implementing IDH inhibitors for the treatment of glioma as well.

Ivosidenib (AG-120), an IDH1 inhibitor, was tested in IDH1-mutant (mIDH1) solid tumors including glioma in a multicenter, open-label, phase I clinical study [51]. Among the 168 patients with solid tumors enrolled in the study, 66 had progressive glioma that recurred or did not respond to standard of care surgery, radiation therapy, or chemotherapy. Patients were classified according to the WHO 2007 classification and *IDH1* mutation status [52]. Glioma patients were separated into two cohorts based on whether the tumor demonstrated contrast enhancement on MRI imaging. The non-enhancing glioma cohort was comprised of patients who experienced tumor progression evidenced by progressive increase in tumor size on at least 3 sets of pre-treatment MRIs completed at least 2 months apart. Of the 66 participants with glioma, 20 were treated in the dose-escalation phase of the trial. No dose-limiting toxicities (DLTs) were observed. Ivosidenib 500 mg once daily (QD) was selected for expansion based on the data from all solid tumors [53]. Treatment-related adverse events (AEs) occurred in 59.1% (39/66) of patients, mostly grade 1/2 fatigue, neutropenia, or diarrhea. Two patients had grade ≥ 3 treatment-related AEs (neutropenia, weight loss, arthralgia, hyponatremia). In the glioma cohort, 35 patients had non-enhancing glioma and 31 patients had enhancing glioma. Median PFS was 13.6 months (95% CI, 9.2 to 33.2 months) for patients with non-enhancing tumors and 1.4 months (95% CI, 1.0 to 1.9 months) for patients with enhancing tumors. Patients with non-enhancing tumors had a median treatment duration of 18.4 months (range, 1.4–47.2 months) compared with a treatment duration of 1.9 months (range, 0.4–39.9 months) for patients with enhancing tumors. Best response of stable disease was observed in 30 of 35 patients with non-enhancing tumors (85.7%) and 14 of 31 patients with enhancing tumors (45.2%) according to the Response Assessment in Neuro-Oncology (RANO) criteria [54,55]. The estimated tumor growth rate per 6 months for the non-enhancing cohort was 26% (95% CI, 9% to 46%) during the pretreatment period, compared to 9% (95% CI, 1% to 20%) during ivosidenib treatment. This study demonstrated ivosidenib was well-tolerated among patients with IDH-mutant glioma and had potentially greater activity against non-enhancing gliomas compared to enhancing gliomas. A potential explanation for this observation is that non-enhancing gliomas generally represent an earlier disease stage with fewer genetic alterations and less chromosomal complexity [56].

Vorasidenib (AG-881), a dual inhibitor of mIDH1/2, was tested in mIDH1/2 solid tumors including glioma in a multicenter, open-label, phase I clinical study [57]. Among 93 mIDH1/2 solid tumors, 52 were gliomas (22 non-enhancing and 30 enhancing tumors). DLT in the form of transaminitis was observed at doses ≥100 mg. A total of 50 mg daily was ultimately determined as the maximum tolerated dose. Treatment-related AEs were reported in 73.1% (38/93) of patients with glioma. The most common grade >3 AE were seizures [4 (7.7%)], elevated alanine aminotransferase (ALT) [3 (5.8%)], and aspartate aminotransferase (AST) levels [2 (3.8%)]. In patients with non-enhancing glioma, the median PFS was 36.8 months (95% CI, 11.2–40.8), compared to 3.6 months (95% CI, 1.8–6.5) among patients with enhancing glioma. The median treatment duration was 26.8 (1.0–50.9) months for patients with non-enhancing glioma and 3.3 (0.2–53.6) months for patients with enhancing glioma. Sixteen (72.7%) patients with non-enhancing glioma had stable disease as their best response. No patients with enhancing glioma had a confirmed radiographic response and 17 of 30 (56.7%) had stable disease as their best response. Overall, vorasidenib was well-tolerated among patients with IDH-mutant glioma with potentially greater disease activity for patients with non-enhancing tumors compared to patients with enhancing tumors. [58]. Inefficacy of vorasidenib in treatment of enhancing glioma was hypothesized to be related to the presence of additional genetic alterations in those tumors with pathways other than gliomagenesis related to the IDH mutation supporting tumor maintenance and growth.

With both ivosidenib and vorasidenib demonstrating evidence of activity in progressive non-enhancing glioma, a randomized phase I perioperative comparative study was subsequently conducted to determine which of the two agents should proceed to phase IIItesting in mIDH glioma. Here, patients with recurrent mIDH glioma were randomized in a 2:2:1 fashion to either the vorasidenib or ivosidenib arm or the untreated arm before surgery [59]. Following the surgery, patients in the untreated arm were re-randomized 1:1 to either vorasidenib or ivosidenib. Tumor 2HG and drug concentrations as well as IDH-pathway related molecular and cellular changes were compared between on-treatment tissue samples and previous surgery tissue samples whenever possible. A total of 24 patients received at least one dose of vorasidenib and 25 patients received at least one dose of ivosidenib. Reduction in mean tumor 2HG levels was greatest with vorasidenib 50 mg daily, compared to low-dose vorasidenib and either dose level of ivosidenib included in the clinical trial. The objective response rate (ORR) was 42.9% (95% CI, 17.7−71.1) for vorasidenib 50 mg daily and 35.7% (95% CI, 12.8−64.9) for ivosidenib 500 mg daily. The median postoperative treatment durations were similar between drugs. Due to superior reduction in tumor 2HG levels with vorasidenib, this mIDH1/2 was selected for a phase III clinical trial. Importantly, evaluation of IDH-pathway related molecular and cellular changes demonstrated that 2HG reduction was associated with reduced tumor cell proliferation, increased DNA 5-hydroxy-methylcitosine content (mediated by TET 5-methylcitosine hydroxylase activity), reversal of gene expression programs of IDH-mutant glioma, induction of genes associated with antitumor immunity, and increase in tumor infiltration with CD8+ T cells. These changes might imply future potential combinatorial targets.

Other IDH inhibitors have also been under investigation for treatment of IDH-mutant glioma. In preclinical studies, the IDH1 inhibitor BAY 1436032 reduced 2HG levels effectively and was linked to reduced cell proliferation and induced glial differentiation, consequently resulting in survival benefit in animal models [60,61]. Accordingly, BAY 1436032 was tested in IDH1-mutant solid tumors including glioma with a multicenter, open-label, phase I clinical study [62]. Among 81 patients, 55 had progressive glioma, with at least one measurable target lesion for the expansion cohort. While near complete (98%) target inhibition was achieved with BAY 1436032 at dosages > 1200 mg, both the 1200 and 1500 mg twice daily doses of the compound were safe and tolerable. Due to the dose-dependent nature of serum 2HG suppression, a dosage of BAY 1436032 1500 mg twice a day was selected for the dose expansion phase. No DLTs were observed during dose escalation phase and a grade 3 maculopapular rash was observed as a DLT during dose expansion phase. Treatment-related adverse events (TRAEs) were reported in 33% of patients, with only one patient experiencing a grade 4 TRAE of elevated lipases. Of 55 patients, 39 had lower grade gliomas, and 16 had secondary glioblastomas, according to the WHO 2016 classification [63]. A total of 35 of 39 patients with IDH-mutant astrocytoma or oligodendroglioma were evaluable via RANO criteria. A best response of stable disease was achieved by 15/35 (43%), and ORR was 4/35 (11%). Five patients achieved a treatment duration longer than 6 months. A total of 14 of 16 patients with IDH-mutant glioblastoma were evaluable via RANO. There were no objective responses, and the best response of stable disease was achieved by 29% (4/14). The efficacy of BAY1436032 among patients with non-enhancing glioma has not been well studied. 

Another IDH1 inhibitor studied for treatment of IDH-mutant glioma is olitasidenib. In a multicenter, open-label, nonrandomized phase Ib/II clinical trial, 26 patients with glioma that had relapsed or that was refractory to standard therapy were treated [64]. Twenty-three patients (88%) who participated in the study had enhancing glioma. Fifteen patients (58%) had grade 3 tumors, whereas 7 (27%) had grade 4 tumors according to the WHO 2016 classification [63]. The median duration of olutasidenib treatment was 4.2 months (1.5–15.2). Twenty-three patients (88%) reported TRAEs. One patient had treatment-related acute hepatitis, and three patients had grade 3 treatment-related transaminitis. The median PFS was 1.9 months (95% CI 1.8–4.6) for the whole study group, whereas the median PFS was 16.9 (95% CI −0.9 to 27.1) months for patients with low-grade gliomas (n = 4). Only 2 of 25 response-evaluable patients (8%) (95% CI 1.0–26.0%) demonstrated an objective response. The disease control rate (objective response plus stable disease) was 48% (12/25 patients). A best response of stable disease was observed in 10/25 (40%). A phase IItrial of olutasidenib for patients with IDH1-mutant pediatric-type high-grade glioma is ongoing (NCT06161974) [65].

An IDH1 inhibitor, DS-1001, has been linked to compromised tumor growth, reduced levels of 2HG, and enhanced expression of GFAP in an orthotopic IDH-mutant glial tumor model [66]. In a multicenter, phase I, open-label clinical trial, DS-1001 was studied in the setting of recurrent or progressive grade 2–4 IDH-mutant glioma [67]. Among 47 patients with glioma, 35 had enhancing, whereas 12 had non-enhancing tumors. One DLT was observed (grade 3 leukopenia). Forty-three percent of patients had at least one grade 3 TRAE with transaminitis and neutropenia being the most common. The ORR was 17.1% for patients with enhancing tumors and 33.3% for patients with non-enhancing tumors. The median treatment duration was 1.7 months for patients with enhancing tumors and 21.3 months for patients with non-enhancing tumors. PFS was 2.4 months for patients with enhancing tumors and not reached for patients with non-enhancing gliomas. Among patients with measurable disease at baseline, 15/35 with enhancing tumors and 11/12 with non-enhancing tumors experienced a decrease in tumor volume. Two patients with enhancing tumors experienced a complete response, whereas three patients with enhancing tumors experienced a partial response. Seven patients were given DS-1001 in an exploratory arm before salvage surgery. Tissue drug concentrations were high and 2HG levels were notably lower than those in archived tissue samples from their previous operations, suggesting mechanisms other than 2HG-mediated gliomagenesis contributed to disease progression in those patients. An open-label, randomized, phase II clinical trial evaluating DS-1001 for IDH-mutant grade 2 and grade 3 progressive gliomas is ongoing (NCT05303519) [68]. In a separate study, DS-1001 is being studied for patients with treatment-naive grade 2 IDH1-mutant glioma (NCT04458272) [69].

Phase I clinical trials of IDH1 inhibitor LY3410738 and an IDH1/2 inhibitor HMPL-306 are ongoing (NCT04521686, NCT04762602) [70,71]. Table 1 summarizes results of early phase glioma studies with IDH inhibitors.

### 3.2. Phase 3 Clinical Study of Vorasidenib in Glioma (The INDIGO Trial)

The efficacy of vorasidenib in residual or recurrent IDH-mutant grade 2 oligodendroglioma or astrocytoma patients was evaluated in a subsequent double-blind, randomized, placebo-controlled phase III clinical trial [72]. Patients were randomly assigned in 1:1 to either vorasidenib 40 mg/day or a placebo. To be eligible, patients had to be within 1 to 5 years of previous surgery for glioma resection before randomization. Measurable disease, defined as a non-enhancing tumor >1 cm, had to be present. High-risk features (uncontrolled seizures, brain-stem involvement, and clinically relevant functional or neurocognitive deficits caused by the tumor) had to be absent. The primary endpoint was PFS, and the secondary endpoints were time to next intervention, ORR, tumor growth rate (TGR), health-related quality of life (HRQoL), and OS.

A total of 331 patients were enrolled, 168 in the vorasidenib arm and 163 in the placebo arm. Imaging-based progression, assessed by blinded independent review, was observed in 47 of 168 patients (28.0%) in the vorasidenib group and in 88 of 163 patients (54.0%) in the placebo group. In the vorasidenib group, the median PFS was 27.7 months (95% CI, 17.0 to not estimated), compared to 11.1 months (95% CI, 11.0 to 13.7) in the placebo group (*p* < 0.001). In the vorasidenib group, 19 patients (11.3%) received another anticancer therapy, whereas in the placebo group, 58 patients (35.6%) received another anticancer intervention. For 52 of these 58 patients, the next intervention represented crossover to vorasidenib. The likelihood of being alive and not receiving a subsequent treatment intervention by 18 months was 85.6% (95% CI, 77.8 to 90.8) in the vorasidenib group, as compared with 47.4% (95% CI, 35.8 to 58.2) in the placebo group (*p* < 0.001). TGRs before and after vorasidenib therapy (n = 56) were 13.2% (95% CI, 10.3, 16.3) and −3.3% (95% CI, −5.2, −1.2) among patients with available imaging data. Vorasidenib was associated with reduced TGR and decreased tumor volume compared to the placebo [73]. HRQoL was better maintained with vorasidenib [74]. The study is ongoing, with reports of other secondary endpoints forthcoming [75].

Subgroup analyses of PFS and time to next intervention favored vorasidenib across most of the subgroups. The most favorable subgroups for both PFS and time to next intervention were tumors with 1p/19q co-deletion and tumors with the longest diameter of more than 2 cm (Hazard ratio for PFS (95% CI): 1p/19q co-deletion status, 0.32 (0.18–0.57) longest diameter of tumor at baseline ≥ 2 cm 0.32 (0.21–0.48). Hazard ratio for next intervention (95% CI): 1p/19q co-deletion status, 0.14 (0.05–0.40); longest diameter of tumor at baseline ≥ 2 cm 0.21 (0.12–0.38).)

Serious AEs occurred in 1.8% of patients in the vorasidenib arm, the most common grade 3 or higher AE being elevated ALT levels. Following initial analysis, the trial was unblinded, and all patients receiving the placebo had the opportunity to crossover to vorasidenib treatment. An expanded access program to provide vorasidenib for IDH-mutant glioma patients was initiated (NCT05592743) [76]. Based on the promising results of the placebo-controlled double-blinded phase III clinical trial, FDA approval of vorasidenib for IDH-mutant glioma is anticipated [77,78]. Development of vorasidenib for treatment of IDH-mutant glioma is illustrated in Figure 2.

### 3.3. IDH Inhibitors Combined with Immunotherapy in Glioma

The IDH mutation and its oncometabolite 2HG interfere with the tumor microenvironment (TME) through several mechanisms. Immune cell migration to TME is compromised in IDH-mutant gliomas compared to IDH-wildtype due to down-regulation of chemotaxis-associated genes, leading to significantly lower tumor-infiltrating lymphocytes (TILs) [79]. Similarly, genes associated with natural killer (NK) cell ligands were silenced in IDH-mutant glioma, allowing the glioma cells to escape from NK-cell surveillance [80]. A comprehensive RNA-sequencing analysis involving 1008 patients with glioma revealed a correlation between mutated IDH and an immune system-related gene signature in glioma. This correlation led to a significantly diminished immune response in IDH-mutant glioma compared to IDH-wildtype [81]. Evidence suggests that 2HG also acts as a paracrine oncometabolite, as T-cells are capable of efficiently taking up 2HG in vitro, leading to impaired T cell activation [82]. The expression of programmed death ligand 1 (PD-L1) was reduced in IDH-mutant gliomas as a result of methylation in the promoter region of the PD-L1 gene [83]. Collectively, mutated IDH can influence several aspects of TME, contributing to immune escape mechanisms. Inhibition of mutated IDH may mitigate its immunosuppressive effects on TME. Therefore, combined use of IDH inhibitors with immunotherapies might enhance their effectiveness through a synergistic effect [84]. In a preclinical study, the combination of a PD-L1 inhibitor and an IDH inhibitor with radiotherapy and temozolomide was shown to be superior to any individual treatment. Clinical trial NCT04056910 is evaluating ivosidenib in combination with nivolumab for patients with IDH-mutant tumors, including glioma. NCT05484622 is evaluating vorasidenib in combination with pembrolizumab (Table 2) [85,86].

In glioma, the most common IDH mutation is IDH1-R132H [4]. A peptide vaccine targeting IDH1 R132H was demonstrated to be safe and effective in inducing antitumor T cell responses [87]. Several other vaccine trials targeting IDH mutation in glioma are ongoing [88,89,90].

## 4. Future Perspectives

The INDIGO trial specifically focused on patients with grade 2 IDH-mutant astrocytomas and oligodendrogliomas, demonstrating early evidence of efficacy with increased PFS compared to the placebo. However, trial data is maturing and the OS benefit for vorasidenib has not yet been confirmed in this patient population. Importantly, 31.9% of patients in the control arm crossed over to the vorasidenib arm after the second interim analysis, which will need to be considered during future survival analysis [91]. Given the relatively long-term survival associated with these low-grade malignancies, survival data will take years to mature.

Notably, it is uncertain if patients with grade 2 tumors who experience progression after vorasidenib treatment would derive similar benefit from radiation therapy and chemotherapy compared to patients who have not had first-line IDH inhibitor treatment. Furthermore, the role of upfront IDH inhibitor treatment for patients with high-risk, low-grade oligodendroglioma or astrocytoma who would be traditionally considered for radiation therapy and chemotherapy following surgery remains unclear. Thus, it is essential to assess the combined and/or sequential use of vorasidenib and chemoradiotherapy in IDH-mutated gliomas. In an in vivo orthotopic IDH-mutant glioma model, researchers demonstrated that radiotherapy and vorasidenib act synergistically, which requires further study in humans [92]. The acquired resistance mechanisms of ivosidenib and vorasidenib are also unknown and need to be addressed, which might include but are not limited to secondary IDH mutations, additional oncogene mutations, and clonal selection [93].

While the INDIGO study offers compelling evidence of the efficacy of vorasidenib for patients with non-enhancing low-grade IDH-mutant glial tumors, the role of vorasidenib in enhancing IDH-mutant astrocytomas and oligodendrogliomas remains unanswered. The data from both phase I studies of vorasidenib and ivosidenib demonstrated that the non-enhancing tumor population had greater PFS than the enhancing tumor population. Also, both studies demonstrated that non-enhancing tumors tend to have more tumor shrinkage than enhancing ones [56,57,58]. Non-enhancing tumors likely represent an earlier disease stage with fewer genetic alterations, more susceptible to IDH inhibition compared to enhancing tumors potentially harboring additional genetic complexity. However, contrast enhancement itself is a prognostic factor in glioma and associated with significantly decreased rates of PFS with different growth rates [94]. In the phase I study of DS-1001, two patients with enhancing tumors did experience a complete response, suggesting there may be a subgroup of patients who would benefit from IDH inhibitors in the setting of enhancing tumors. [67]. Combinatorial strategies for IDH inhibitors and conventional treatments such as radiation therapy and alkylator chemotherapy for patients with high-grade IDH-mutant astrocytomas and oligodendrogliomas require further study.

The perioperative study of vorasidenib and ivosidenib demonstrated that patients with abnormal cell cycle genes have shorter PFS independent from contrast enhancement status [59]. Early phase studies to date suggest potential limited benefit of IDH inhibitor monotherapy in the setting of higher-grade tumors. However, distinguishing grade 2 from grade 3 tumors is not always straightforward. Indeed, a number of studies have demonstrated that the disparities in survival rates between IDH-mutant tumors of grades 2 and 3 are, in fact, quite limited [95]. In the era of IDH inhibitors, the question of what precisely distinguishes an IDH-mutant CNS WHO grade 2 glioma from a grade 3 glioma may become increasingly relevant for treatment decision-making [96]. Techniques to improve the accuracy of tumor grading of diffuse gliomas—for example, DNA methylation profiling or addition of molecular markers—require further investigation and may in time further define patient populations who benefit from administration of IDH inhibitors [97,98].

Combinatorial strategies beyond IDH inhibitor monotherapy are already under investigation. Trials combining ivosidenib with immune checkpoint inhibitors are underway. Other areas of interest include the metabolic vulnerabilities associated with IDH-mutant tumors and DNA damage repair pathways. As an example, given the DNA hypermethylation phenotype driven by 2HG, demethylating agents may achieve tumor control. Currently, 5-azacytidine and ASTX727, a fixed-dose combination of cedazuridine and decitabine, is being tested in IDH-mutated glial tumors (NCT03666559, NCT03922555) [99,100]. Diminished NAD+ levels were associated with elevated DNA damage in IDH-mutant glioma, as poly ADP ribose polymerase (PARP)-mediated DNA repair is dependent on NAD+ levels. Several clinical trials testing PARP inhibitors in IDH-mutated gliomas are ongoing (NCT03212274, NCT03991832, NCT03914742, and NCT05076513) [101,102,103,104]. Combination of these approaches with IDH inhibitors can be considered.

The initial positive results from the INDIGO trial represent a significant step forward for treatment of IDH-mutant glioma. The data so far suggest efficacy of IDH inhibitors in the setting of low-grade, non-enhancing glioma. Upon approval, vorasidenib is likely to be broadly used as first-line treatment for low-grade glioma, representing a strategy to delay radiation therapy and chemotherapy until first progression. Whether upfront use of IDH inhibitors is associated with an overall survival benefit remains unclear and represents a critical question that will be answered with maturation of INDIGO data. As these novel therapeutics enter clinical practice, we will also need to determine whether there is a role for the combination of IDH inhibitors with conventional treatments. Elucidating the role of IDH inhibitors as maintenance therapy following completion of radiation therapy and alkylator chemotherapy represents another area of interest. Upon disease progression, identification of resistance mechanisms to IDH inhibitors may uncover additional vulnerabilities exploited by novel therapeutics, similar to the development of multiple generations of EGFR and ALK inhibitors currently available for clinical use. Despite current unknowns, encouraging results from early phase IDH inhibitor studies and the phase III vorasidenib trial underscore that there is a clear and important role for precision oncology in brain tumor care.

## 5. Conclusions

IDH inhibitors are becoming increasingly a part of clinical practice in neuro-oncology. Initial results of the phase III clinical trial with vorasidenib are highly encouraging, representing a significant advancement in IDH-mutant glioma treatment. Ongoing and future studies will further define groups of patients most likely to benefit from IDH inhibitors, either as monotherapy or as part of combinatorial strategies.

## Figures and Tables

**Figure 1 pharmaceuticals-17-00682-f001:**
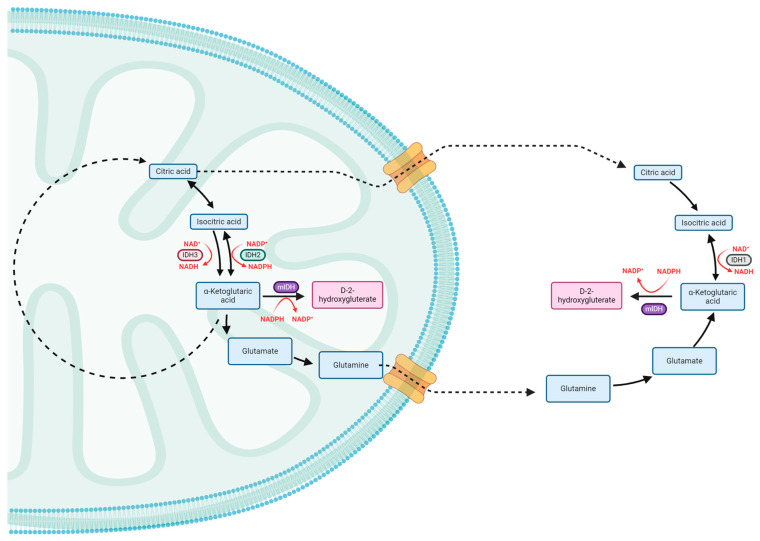
Physiological functions of isocitrate dehydrogenase isoforms in normal cellular metabolisms are demonstrated in black. The aberrant activity of mutated isocitrate dehydrogenase isoforms and its results are demonstrated in red.

**Figure 2 pharmaceuticals-17-00682-f002:**
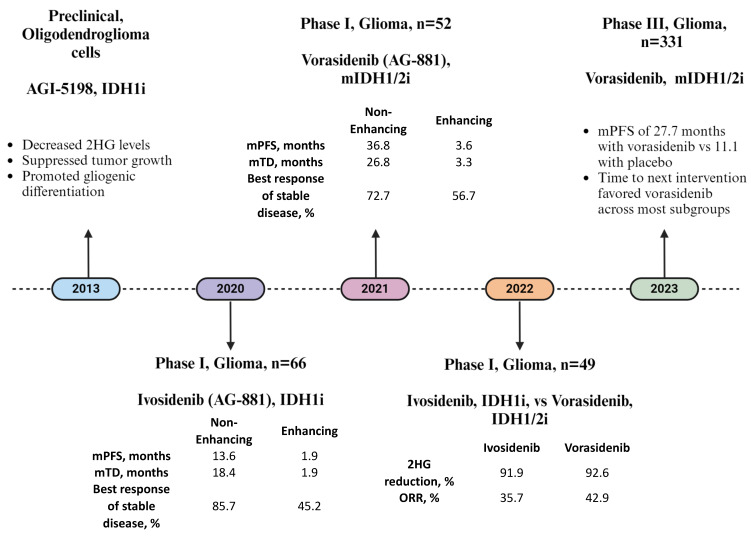
The timeline of events leading to availability of ivosidenib in clinical practice in neuro-oncology is summarized.

**Table 1 pharmaceuticals-17-00682-t001:** Comparison of the design and results of the early phase clinical trials of IDH inhibitors in glioma.

Title	Study Type	Drug Name	Target	Escalation Range	Expansion Dose	Eligibility Criteria	No. of Patients	Safety	Study Cohort	mPFS, Months	BRSD, %	mTD, Months	ORR
Ivosidenib in Isocitrate Dehydrogenase 1–Mutated Advanced Glioma	multicenter, open-label, phase I clinical study	ivosidenib	mutant IDH1	100–900 mg once daily	500 mg once daily	Progressive glioma	66	No DLTs TRAE 59.1% Gr > 3 TRAE in 2 patients	Non-enhancing: 35 vs. enhancing: 31 12/66 GBM *	13.6 vs. 1.4 mo	85.7% vs. 45.2%	18.4 vs. 1.9	2.9% vs. 0%
Vorasidenib, a Dual Inhibitor of Mutant IDH1/2, in Recurrent or Progressive Glioma; Results of a First-in-Human Phase I Trial	multicenter, open-label, phase I clinical study	vorasidenib	mIDH1/2	25–300 mg once daily	50 mg once daily	Progressive glioma	52	DLTs of elevated ALT/AST TRAE 73.1% Gr > 3 AE in 10 patients	Non-enhancing: 22 vs. enhancing: 30	36.8 vs. 3.6	72.7% vs. 56.7%	26.8 vs. 3.3	18% vs. 0%
Vorasidenib and ivosidenib in IDH1-mutant low-grade glioma: a randomized, perioperative phase 1 trial	randomized phase I perioperative comparative study	Ivosidenib vs. vorasidenib	-	Ivosidenib 250 mg or 500 mg daily, Vorasidenib 10 mg or 50 mg daily	Vorasidenib 50 mg daily	Recurrent glioma before surgery	49	Gr >3 AE 24% with ivosidenib vs. 29.2% with vorasidenib	Ivosidenib: 25 vs. vorasidenib: 24	-	-	15.1 vs. 14.3	35.7% vs 42.9%
Phase I assessment of safety and therapeutic activity of BAY1436032 in patients with IDH1-mutant solid tumors	multicenter, open-label, phase I clinical study	BAY 1436032	IDH1 (pan-inhibitor)	150–1500 mg twice daily	1500 mg twice daily	Progressive glioma with a target lesion	55	DLT of Gr 3 maculopapular rash TRAE in 33% Gr 4 TRAE of elevated lipase	33/35 enhancing LGG, 14 GBM **	-	43%	-	15%
Olutasidenib (FT-2102) in patients with relapsed or refractory IDH1-mutant glioma: A multicenter, open label, phase Ib/II trial	multicenter, open-label, nonrandomized phase Ib/II clinical study	Olutasidenib	IDH1	-	150 mg twice daily	Progressive glioma	26	TRAE 88% Gr 3 TRAE in 3 patients (elevated ALT/AST)	23/26 enhancing, 4/26 LGG, 15/26 Grade III, 7/26 Grade 4	1.9 vs. 16.9 ***	40%	-	8%
The first-in-human phase I study of a brain-penetrant mutant IDH1 inhibitor DS-1001 in patients with recurrent or progressive IDH1-mutant gliomas	multicenter, open-label, phase I clinical study	DS-1001	IDH1	125–1400 mg twice daily	250 mg twice daily	Progressive glioma	47	DLT of Gr 3 WBC decrease Gr 3 TRAE 42.6%	35 enhancing vs. 12 non-enhancing	not-reached vs. 2.4	66.7% vs. 31.4%	21.3 vs. 1.7	33.3% vs. 17.1%

IDH, isocitrate dehydrogenase; DLT, dose limiting toxicities; LGG, low grade glioma; TRAE, treatment-related adverse events; ALT, alanine transaminase; AST, aspartate transaminase; Gr, grade; AE, adverse event; GBM, glioblastoma; No, number; mPFS, median progression-free survival; mTD, median treatment duration; ORR, overall response rate; BRSD, best response of stable disease. * according to the WHO 2007 classification. ** 35/39 LGG and 14/16 GBM were evaluable via RANO. IDH-mutant GBM according to the WHO classification 2016 edition. *** 1.9 months for whole study group vs. 16.9 months for LGG patients.

**Table 2 pharmaceuticals-17-00682-t002:** Ongoing clinical trials of IDH inhibitors in glioma.

Study Name	NCT Number	Drug Name	Target	Phase	Eligibility Criteria	No. of Patients
Study of Olutasidenib and Temozolomide in HGG	NCT06161974	olutasidenib	IDH1	II	pediatric and young adult patients newly diagnosed with a high-grade glioma (HGG) that have a genetic mutation in IDH1	65
Safusidenib Phase 2 Study in IDH1 Mutant Glioma	NCT05303519	safusidenib	IDH1	II	recurrent or progressive histologically confirmed IDH1 mutant WHO grade 2/3 glioma	95
A Study of DS-1001b in Patients with Chemotherapy- and Radiotherapy-Naive IDH1 Mutated WHO Grade II Glioma	NCT04458272	DS-1001b	IDH1	II	chemotherapy- and radiotherapy-naive IDH1 mutated WHO grade 2 glioma	25
Study of LY3410738 Administered to Patients With Advanced Solid Tumors With IDH1 or IDH2 Mutations	NCT04521686	LY3410738	IDH1	I	IDH1 R132-mutant advanced solid tumors, including glioma	NA
A Study of HMPL-306 in Advanced Solid Tumors With IDH Mutations	NCT04762602	HMPL-306	dual IDH1/2	I	solid tumors including low-grade glioma, perioperative low-grade glioma	NA
Vorasidenib Expanded Access Program	NCT05592743	Vorasidenib	dual IDH1/2	Expanded access	IDH1- or IDH2-mutated glioma	NA
Ivosidenib (AG-120) With Nivolumab in IDH1 Mutant Tumors	NCT04056910	Ivosidenib + Nivolumab	IDH1 + PD-1	II	advanced solid tumors (nonresectable or metastatic) or enhancing gliomas	NA
Study of Vorasidenib and Pembrolizumab Combination in Recurrent or Progressive Enhancing IDH-1 Mutant Astrocytomas	NCT05484622	Vorasidenib + Pembrolizumab	Dual IDH1/2 + PD-1	I	recurrent or progressive enhancing isocitrate dehydrogenase-1 (IDH-1) mutant astrocytomas	72

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
