# Peer review of "Isocitrate Dehydrogenase Inhibitors in Glioma: From Bench to Bedside"

_pharmaceuticals, 2024, doi:10.3390/ph17060682_

Round 1

Reviewer 1 Report

Comments and Suggestions for Authors

In this review, the authors have summarized an overview of the ongoing clinical trials investigating IDH mutant inhibitors. They have outlined the patient cohorts, treatment specifics, and the preliminary findings obtained thus far. Overall, the review is well organized and serves as a comprehensive summary of the current landscape of clinical trials focusing on IDH mutant inhibitors. However, there are a few points that require attention to enhance the manuscript further.

1. The introduction section would benefit from further elaboration on the epigenetic regulations induced by IDH mutation. Specifically, it's important to include how IDH1 mutation leads to alterations in chromatin structure by impeding CTCF binding through hypermethylation. Several relevant studies can be cited to support these points, eg, Insulator dysfunction and oncogene activation in IDH mutant gliomas 2016 Nature; Modeling epigenetic lesions that cause gliomas, 2023, Cell.

2. It would be helpful if the authors could add up the research on the effects of IDH mutations on KDM4 family and TET protein functions in gliomas.

3. Figure 1. it would be better if the authors mark up the mitochondria and cytoplasm sections. And a larger font could be helpful for reading.

4. Line 115, the reference for this study is missing.

5. The clinical trials investigating the combination treatment of TMZ or other alkylating agents with IDHmut inhibitors could be consolidated into a single section since alkylating agents are currently the primary first-line drugs for glioma treatment.

Author Response

05.16.2024

To the Editor of Pharmaceuticals,

We would like to thank you for the comments on the manuscript ‘‘Isocitrate Dehydrogenase Inhibitors in Glioma: From Bench to Bedside’’. The necessary changes based on the suggestions of the reviewers have been made. Please find our responses to the comments below.

Thank you for your contributions.

Best regards,

Merve Hazal Ser, M.D.

Review Report Form 1

Comments and Suggestions for Authors

In this review, the authors have summarized an overview of the ongoing clinical trials investigating IDH mutant inhibitors. They have outlined the patient cohorts, treatment specifics, and the preliminary findings obtained thus far. Overall, the review is well organized and serves as a comprehensive summary of the current landscape of clinical trials focusing on IDH mutant inhibitors. However, there are a few points that require attention to enhance the manuscript further.

  1. The introduction section would benefit from further elaboration on the epigenetic regulations induced by IDH mutation. Specifically, it's important to include how IDH1 mutation leads to alterations in chromatin structure by impeding CTCF binding through hypermethylation. Several relevant studies can be cited to support these points, eg, Insulator dysfunction and oncogene activation in IDH mutant gliomas 2016 Nature; Modeling epigenetic lesions that cause gliomas, 2023, Cell.

Thank you. A paragraph is added with relevant references.

‘DNA hypermethylation is driven by 2HG, which blocks the activity of the DNA de-methylation enzyme and a glioma-specific DNA methylation pattern occurs over time, leading to transcriptional silencing of tumor suppressor genes [24]. Ten-eleven translo-cation (TET) 5′ methylcytosine hydroxylase enzyme activity, which catalyzes a key step in the removal of DNA methylation, is blocked by 2HG. Thus IDH mutations manifest a CpG island methylator phenotype (G-CIMP), leading gene expression programs of IDH-mutant glioma [25]. In addition, IDH mutations cause hypermethylation of CCCTC binding factor (CTCF) binding sites genome-wide, leading reduced CTCF binding [26]. CTCF insulator protein is responsible for the partition of ‘contact domains’ of the ge-nome, which are discrete structural and regulatory units. Loss of CTCF at a domain boundary is linked to altered gene insulations, which contribute to gliomagenesis by activating oncogenes that are normally insulated. In mouse oligodendrocyte progenitor cells, disruption of CTCF insulator near the platelet-derived growth factor receptor A (PDGFRA), a prominent glioma oncogene, demonstrated to be related with increased proliferation [27].’

  1. It would be helpful if the authors could add up the research on the effects of IDH mutations on KDM4 family and TET protein functions in gliomas. 

Thank you. This sentence is revised:

‘Histone lysine demethylases (KDMs) are also inhibited by 2HG, which compromises cellular differentiation and contributes to the glioma cells' regressing to a more “primi-tive” developmental state [26]. Histone methylation by KDM inhibiton contributes to oncogenesis in IDH-mutant tumors through glial transformation in IDH-mutant cells [27, 28].’

  1. Figure 1. it would be better if the authors mark up the mitochondria and cytoplasm sections. And a larger font could be helpful for reading.

Thank you. This figure is updated.

  1. Line 115, the reference for this study is missing.

Thank you. It is added.

  1. The clinical trials investigating the combination treatment of TMZ or other alkylating agents with IDHmut inhibitors could be consolidated into a single section since alkylating agents are currently the primary first-line drugs for glioma treatment.

Thank you for your comment. However, there is only one clinical trial combining an alkylating agent -temozolomide- with an IDH inhibitor; therefore there is not enough trials to create a section.

Reviewer 2 Report

Comments and Suggestions for Authors

The authors present a Review manuscript describing the use of Isocitrate Dehydrogenase inhibitors in Glioma in a comprehensive view; “from the bench to the bedside”.

The title is short presenting the information in a clear manner.

The Abstract describes the argument and contextualises the use of isocitrate dehydrogenase (IDH) inhibitors on the glioma landscape. The authors discuss the importance of IDH mutations in oncogenesis of adult-type diffuse gliomas and the current evidence supporting using IDH inhibitors as therapeutic agents for glioma treatment. We also examine unresolved questions and propose potential directions for future research.

Overall, the manuscript 

The introduction has the correct and expected size and is presented clearly and linearly. Authors present the state-of-the-art information about brain tumors. The Introduction summarizes the need of novel treatments that would provide patients better quality of life, survival and would mitigate the toxicities associated with conventional therapies. 

Information regarding lines 44 and 45 “More recently, IDH has emerged as a treatment target with promising clinical trial results 44 now establishing the role of IDH inhibitors in glioma treatment” is misleading. Please re-type the sentence. 

In this section, authors should reconsider to reformat the references, e.g. [2] [3] [4] [5] on line 38 should be avoided; it should present as [2-5]. Same, still in the introduction section is present in line 55. Please re-format all the manuscript regarding to this issue.

Authors, in the correct way, finished the introduction section by mentioning the aim of the present review manuscript which can help readers to decide whether to keep on with the reading or not.

Lines 79-80 should be contextualised in a better way. Information before these lines is not enough to understand why “the oncometabolite causes a reduction of glucolysis…, …non-Krebs cycle sources of carbohydrates are needed”. Please add information on this, otherwise is difficult to understand. 

In line 94, the acronym ROS appears without mentioning its meaning. Please add it before the acronym.

In page 5, the sentence on lanes 215-16 “BAY 1436032 150 mg twice …were safe and tolerable”, is misleading. Please re-type it for better understanding.

Due to the type of manuscript, the document presents a large number of acronyms, and for this, ALL of them should be listed at the beginning of the manuscript. Please add a full list of acronyms used in the paper. This will guide readers throughout the paper.

Authors present a long “Future Perspectives” section where explanations and personal scientific opinions, based on referred manuscripts and clinical trials, on IDH and glioma treatment are discussed. This section of the manuscript is clear and the insertion of Clinical Trials references, both with the acronyms and reference numbers demonstrate scientific consistency and care.  The section is linear, clear and easy to comprehend. The conclusion section closes the manuscript in a very simplistic way. Authors remark ongoing and future landscape of IDH inhibitors in the neuro-oncological field in a short way. Please include a personal view of future direction to be taken.

Figures and Tables

The authors present an illustration of the physiological functions of IDH isoforms in Figure 1.  The figure is well-designed and can represent, in a simplistic manner, some cellular metabolic events involving IDH. Authors present a colour code – aberrant activity is represented in red – that is of easy understanding. Despite this, the figure lacks quality, in particular intracellular mechanisms and pathways. Please increase it. It would be a PLUS for the manuscript.

Figure 2 illustrates the timeline of events during a clinical practice. The figure is well represented but lacks quality, especially on the tables. Please increase it for better quality of the manuscript.  

Both tables are well designed and are of clear reading and understanding. Nevertheless, it would be a Plus for the manuscript quality to place the first raw of both tables: Table 1 Comparison of the design and results of the early phase clinical trials of IDH inhibitors in gliomas and Table 2 Ongoing Clinical Trials of IDH Inhibitors in Glioma, on alphabetic order. Please do so. 

Also, on both table legends, it is not necessary to type all the orders in caps look. Please change it.

References, e.g. scientific manuscripts and clinical trial reports, support and correctly sustain the manuscript. As mentioned before, please correct Refs numbers.  This must be changed, details are important for final manuscript quality.

The present manuscript, upon the abovementioned revisions, would be considered for publication and well-suited for the audience of Pharmaceutics special issue. Ultimately, I would be available to re-evaluate the manuscript after the abovementioned adjustments.

Comments on the Quality of English Language

The authors present a Review manuscript describing the use of Isocitrate Dehydrogenase inhibitors in Glioma in a comprehensive view; “from the bench to the bedside”.

The title is short presenting the information in a clear manner.

The Abstract describes the argument and contextualises the use of isocitrate dehydrogenase (IDH) inhibitors on the glioma landscape. The authors discuss the importance of IDH mutations in oncogenesis of adult-type diffuse gliomas and the current evidence supporting using IDH inhibitors as therapeutic agents for glioma treatment. We also examine unresolved questions and propose potential directions for future research.

Overall, the manuscript 

The introduction has the correct and expected size and is presented clearly and linearly. Authors present the state-of-the-art information about brain tumors. The Introduction summarizes the need of novel treatments that would provide patients better quality of life, survival and would mitigate the toxicities associated with conventional therapies. 

Information regarding lines 44 and 45 “More recently, IDH has emerged as a treatment target with promising clinical trial results 44 now establishing the role of IDH inhibitors in glioma treatment” is misleading. Please re-type the sentence. 

In this section, authors should reconsider to reformat the references, e.g. [2] [3] [4] [5] on line 38 should be avoided; it should present as [2-5]. Same, still in the introduction section is present in line 55. Please re-format all the manuscript regarding to this issue.

Authors, in the correct way, finished the introduction section by mentioning the aim of the present review manuscript which can help readers to decide whether to keep on with the reading or not.

Lines 79-80 should be contextualised in a better way. Information before these lines is not enough to understand why “the oncometabolite causes a reduction of glucolysis…, …non-Krebs cycle sources of carbohydrates are needed”. Please add information on this, otherwise is difficult to understand. 

In line 94, the acronym ROS appears without mentioning its meaning. Please add it before the acronym.

In page 5, the sentence on lanes 215-16 “BAY 1436032 150 mg twice …were safe and tolerable”, is misleading. Please re-type it for better understanding.

Due to the type of manuscript, the document presents a large number of acronyms, and for this, ALL of them should be listed at the beginning of the manuscript. Please add a full list of acronyms used in the paper. This will guide readers throughout the paper.

Authors present a long “Future Perspectives” section where explanations and personal scientific opinions, based on referred manuscripts and clinical trials, on IDH and glioma treatment are discussed. This section of the manuscript is clear and the insertion of Clinical Trials references, both with the acronyms and reference numbers demonstrate scientific consistency and care.  The section is linear, clear and easy to comprehend. The conclusion section closes the manuscript in a very simplistic way. Authors remark ongoing and future landscape of IDH inhibitors in the neuro-oncological field in a short way. Please include a personal view of future direction to be taken.

Figures and Tables

The authors present an illustration of the physiological functions of IDH isoforms in Figure 1.  The figure is well-designed and can represent, in a simplistic manner, some cellular metabolic events involving IDH. Authors present a colour code – aberrant activity is represented in red – that is of easy understanding. Despite this, the figure lacks quality, in particular intracellular mechanisms and pathways. Please increase it. It would be a PLUS for the manuscript.

Figure 2 illustrates the timeline of events during a clinical practice. The figure is well represented but lacks quality, especially on the tables. Please increase it for better quality of the manuscript.  

Both tables are well designed and are of clear reading and understanding. Nevertheless, it would be a Plus for the manuscript quality to place the first raw of both tables: Table 1 Comparison of the design and results of the early phase clinical trials of IDH inhibitors in gliomas and Table 2 Ongoing Clinical Trials of IDH Inhibitors in Glioma, on alphabetic order. Please do so. 

Also, on both table legends, it is not necessary to type all the orders in caps look. Please change it.

References, e.g. scientific manuscripts and clinical trial reports, support and correctly sustain the manuscript. As mentioned before, please correct Refs numbers.  This must be changed, details are important for final manuscript quality.

The present manuscript, upon the abovementioned revisions, would be considered for publication and well-suited for the audience of Pharmaceutics special issue. Ultimately, I would be available to re-evaluate the manuscript after the abovementioned adjustments.

Author Response

05.16.2024

To the Editor of Pharmaceuticals,

We would like to thank you for the comments on the manuscript ‘‘Isocitrate Dehydrogenase Inhibitors in Glioma: From Bench to Bedside’’. The necessary changes based on the suggestions of the reviewers have been made. Please find our responses to the comments below.

Thank you for your contributions.

Best regards,

Merve Hazal Ser, M.D.

Review Report Form 2

Comments and Suggestions for Authors

The authors present a Review manuscript describing the use of Isocitrate Dehydrogenase inhibitors in Glioma in a comprehensive view; “from the bench to the bedside”.

The title is short presenting the information in a clear manner.

The Abstract describes the argument and contextualises the use of isocitrate dehydrogenase (IDH) inhibitors on the glioma landscape. The authors discuss the importance of IDH mutations in oncogenesis of adult-type diffuse gliomas and the current evidence supporting using IDH inhibitors as therapeutic agents for glioma treatment. We also examine unresolved questions and propose potential directions for future research.

Overall, the manuscript 

The introduction has the correct and expected size and is presented clearly and linearly. Authors present the state-of-the-art information about brain tumors. The Introduction summarizes the need of novel treatments that would provide patients better quality of life, survival and would mitigate the toxicities associated with conventional therapies. 

Information regarding lines 44 and 45 “More recently, IDH has emerged as a treatment target with promising clinical trial results 44 now establishing the role of IDH inhibitors in glioma treatment” is misleading. Please re-type the sentence. 

Thank you. This sentence is deleted.

In this section, authors should reconsider to reformat the references, e.g. [2] [3] [4] [5] on line 38 should be avoided; it should present as [2-5]. Same, still in the introduction section is present in line 55. Please re-format all the manuscript regarding to this issue.

Thank you. They are organized.

Authors, in the correct way, finished the introduction section by mentioning the aim of the present review manuscript which can help readers to decide whether to keep on with the reading or not.

Thank you for your comment.

Lines 79-80 should be contextualised in a better way. Information before these lines is not enough to understand why “the oncometabolite causes a reduction of glucolysis…, …non-Krebs cycle sources of carbohydrates are needed”. Please add information on this, otherwise is difficult to understand. 

Thank you. These sentences are clarified:

‘Given that αKG serves as the substrate for glycolysis in the Krebs cycle, the oncometabolite 2HG subsequently leads to a reduction in glycolysis by depleting αKG. To compensate, non-Krebs cycle sources of carbohydrates are needed.’

In line 94, the acronym ROS appears without mentioning its meaning. Please add it before the acronym.

Thank you. It is added.

‘The presence of IDH mutation also leads to accumulation of reactive oxygen species (ROS), which are detrimental to DNA, lipids, and proteins.’

In page 5, the sentence on lines 215-16 “BAY 1436032 150 mg twice …were safe and tolerable”, is misleading. Please re-type it for better understanding.

Thank you. It is clarified:

‘While near-complete (98%) target inhibition was achieved with BAY 1436032 at dosages> 1200 mg, both the 1200 and 1500 mg twice daily doses of the compound were safe and tolerable. Due to the dose-dependent nature of serum 2HG suppression, a dosage of BAY 1436032 1500 mg twice a day was selected for the dose expansion phase.’

Due to the type of manuscript, the document presents a large number of acronyms, and for this, ALL of them should be listed at the beginning of the manuscript. Please add a full list of acronyms used in the paper. This will guide readers throughout the paper.

Thank you. Acronyms are listed following the abstract.

Authors present a long “Future Perspectives” section where explanations and personal scientific opinions, based on referred manuscripts and clinical trials, on IDH and glioma treatment are discussed. This section of the manuscript is clear and the insertion of Clinical Trials references, both with the acronyms and reference numbers demonstrate scientific consistency and care.  The section is linear, clear and easy to comprehend. The conclusion section closes the manuscript in a very simplistic way. Authors remark ongoing and future landscape of IDH inhibitors in the neuro-oncological field in a short way. Please include a personal view of future direction to be taken.

Thank you. A paragraph explaining personal view of future direction to be taken is included:

‘The initial positive results from the INDIGO trial represent a significant step forward for treatment of IDH-mutant glioma. The data so far suggests efficacy of IDH inhibitors in the setting of low-grade, non-enhancing glioma. Upon approval, vorasidenib is likely to be broadly used as first-line treatment for low-grade glioma, representing a strategy to delay radiation therapy and chemotherapy until first progression. Whether upfront use of IDH inhibitors is associated with an overall survival benefit remains unclear and represents a critical question that will be answered with maturation of INDIGO data. As these novel therapeutics enter clinical practice, we will also need to determine whether there is a role for combination of IDH inhibitors with conventional treatments. Elucidating the role of IDH inhibitors as maintenance therapy following completion of radiation therapy and alkylator chemotherapy represents another area of interest. Upon disease progression, identification of resistance mechanisms to IDH inhibitors may uncover additional vulnerabilities exploited by novel therapeutics, similar to development of multiple generations of EGFR and ALK inhibitors currently available for clinical use. Despite current unknowns, encouraging results from early phase IDH inhibitor studies and phase 3 vorasidenib trial underscore that there is a clear and important role for precision oncology in brain tumor care.’

Figures and Tables

The authors present an illustration of the physiological functions of IDH isoforms in Figure 1.  The figure is well-designed and can represent, in a simplistic manner, some cellular metabolic events involving IDH. Authors present a colour code – aberrant activity is represented in red – that is of easy understanding. Despite this, the figure lacks quality, in particular intracellular mechanisms and pathways. Please increase it. It would be a PLUS for the manuscript.

Thank you for your suggestions. Figure 1 is updated accordingly. 

Figure 2 illustrates the timeline of events during a clinical practice. The figure is well represented but lacks quality, especially on the tables. Please increase it for better quality of the manuscript.  

Thank you. Figure 2 is changed.

Both tables are well designed and are of clear reading and understanding. Nevertheless, it would be a Plus for the manuscript quality to place the first raw of both tables:  Table 1 Comparison of the design and results of the early phase clinical trials of IDH inhibitors in gliomas and Table 2 Ongoing Clinical Trials of IDH Inhibitors in Glioma, on alphabetic order. Please do so. 

Thank you.

Also, on both table legends, it is not necessary to type all the orders in caps look. Please change it.

Thank you, it is changed.

References, e.g. scientific manuscripts and clinical trial reports, support and correctly sustain the manuscript. As mentioned before, please correct Refs numbers.  This must be changed, details are important for final manuscript quality.

Thank you. They are organized.

The present manuscript, upon the abovementioned revisions, would be considered for publication and well-suited for the audience of Pharmaceutics special issue. Ultimately, I would be available to re-evaluate the manuscript after the abovementioned adjustments.

Thank you very much for your precious comments and scientific contributions.

Reviewer 3 Report

Comments and Suggestions for Authors

This review addresses the topic of therapeutic exploitation of unique targets presented by IDH mutations in glioma and describes published work that supports the utility of this approach.  A PubMed search revealed a number of recent (2023) papers that should be added to this review". Tables and figures are good, but additional recent published work should be included.

Author Response

05.16.2024

To the Editor of Pharmaceuticals,

We would like to thank you for the comments on the manuscript ‘‘Isocitrate Dehydrogenase Inhibitors in Glioma: From Bench to Bedside’’. The necessary changes based on the suggestions of the reviewers have been made. Please find our responses to the comments below.

Thank you for your contributions.

Best regards,

Merve Hazal Ser, M.D.

Review Report Form 3

Comments and Suggestions for Authors

This review addresses the topic of therapeutic exploitation of unique targets presented by IDH mutations in glioma and describes published work that supports the utility of this approach.  A PubMed search revealed a number of recent (2023) papers that should be added to this review". Tables and figures are good, but additional recent published work should be included.

Thank you for your contribution. Several recent papers are included into the references in the revised version.